# Mechanisms Underlying Antipsychotic-Induced NAFLD and Iron Dysregulation: A Multi-Omic Approach

**DOI:** 10.3390/biomedicines10061225

**Published:** 2022-05-24

**Authors:** Meghan May, Deborah Barlow, Radwa Ibrahim, Karen L. Houseknecht

**Affiliations:** Department of Biomedical Sciences, College of Osteopathic Medicine, University of New England, Biddeford, ME 04005, USA; dbarlow@une.edu (D.B.); ribrahim1@une.edu (R.I.)

**Keywords:** NAFLD, NASH, antipsychotic, inflammation, insulin resistance, iron metabolism, anemia, psychiatry

## Abstract

Atypical antipsychotic (AA) medications are widely prescribed for the treatment of psychiatric disorders, including schizophrenia, bipolar disorder and treatment-resistant depression. AA are associated with myriad metabolic and endocrine side effects, including systemic inflammation, weight gain, dyslipidemia and insulin resistance, all of which are associated with increased incidence of non-alcoholic fatty liver disease (NAFLD). NAFLD is highly prevalent in patients with mental illness, and AA have been shown to increase incidence of NAFLD pre-clinically and clinically. However, the underlying mechanisms have not been described. We mined multi-omic datasets from preclinical murine models of sub-chronic risperidone or olanzapine treatment, in vitro exposure of human cells to risperidone and psychiatric patients following onset of aripiprazole therapy focused on pathways associated with the pathophysiology of NAFLD, including iron accumulation, systemic inflammation and dyslipidemia. We identified numerous differentially expressed traits affecting these pathways conserved across study systems and AA medications. We used these findings to propose mechanisms for AA-associated development of NAFLD and dysregulated iron homeostasis.

## 1. Introduction

Non-alcoholic fatty liver disease (NAFLD) is the most common cause of liver disease in the world and is characterized by the presence of steatosis in >5% of hepatocytes by histological analysis [1]. It has a prevalence rate of up to 25–30% in the general population with an upwards trend. Non-alcoholic steatohepatitis (NASH) represents advanced-stage disease progression, leading to cirrhosis and ultimately liver failure, making NASH one of the leading indications for liver transplantation [2]. NAFLD, the hepatic manifestation of insulin resistance and metabolic syndrome, is highly comorbid with obesity, and is considered an independent risk factor for cardiovascular disease [3,4]. Furthermore, growing evidence supports an association between metabolic syndrome and psychiatric disorders, such as schizophrenia, bipolar disorder and depression [5]. A recent study examining psychiatric patients without prior liver-related events found that liver disease was the fifth most common cause of death in this population [6]. Myriad factors likely contribute to increased NAFLD incidence in patients with mental illness, including the effects of psychiatric medications such as antipsychotic drugs [6,7,8]. Antipsychotic medications have complex and diverse pharmacology and are prescribed for myriad indications including schizophrenia, bipolar disorder, treatment resistant depression and irritability associated with autism [9]. These drugs are associated with significant endocrine, metabolic and inflammatory side effects including insulin resistance, dyslipidemia and NAFLD, and drug effects are distinct from metabolic side effects associated with psychiatric illness [10,11]. Rapid weight gain is a common occurrence in first-episode psychosis treated with AA medications [12,13,14], and the majority of patients have persistent body mass indices (BMI) classified as obese [15]. We and others have reported that clinically relevant exposures of the atypical antipsychotic (AA) medications risperidone and olanzapine induced histologically confirmed NAFLD with coincident alteration in proteomic signatures consistent with insulin resistance, dysregulated inflammation and mitochondrial dysfunction in pre-clinical models [7,9,16,17].

Development of NAFLD is complex and multifaceted (Figure 1), and some of the underlying pathophysiological mechanisms are not well understood. One such mechanism is the relationship between NAFLD and iron homeostasis. Patients with NAFLD often present with alterations in iron metabolism, and there is an increase in hepatic iron stores in about one third of patients with NAFLD [18,19].

Dysregulation of iron metabolism is associated with cognitive disorders, including memory and attention disorders, with or without anemia [20,21,22]. Although the molecular mechanisms are not well defined, iron overload induces insulin resistance and increases oxidative stress, which in turn leads to further alteration in insulin signaling pathways and fat metabolism [23]. Antipsychotic medications can also disrupt iron metabolism, leading to akathisia and hypoferritinemia/iron-deficiency anemia, particularly in pediatric patients [24,25,26].

The association of NAFLD with disrupted iron homeostasis, coupled with the induction of both states by AA medications, suggests a complex, interdependent relationship between AA-induced NAFLD and dysregulated iron metabolism. In this study we used a multi-omic translational approach to evaluate AA-induced changes in pathways linked to NAFLD and iron metabolism. We identified differentially expressed (DE) traits conserved across study systems including three AA medications and three treatment modalities and used these findings to propose pathophysiological mechanisms of AA-induced NAFLD with disrupted iron homeostasis.

## 2. Materials and Methods

### 2.1. Proteomic and Immunomic Changes Induced by Risperidone and Olanzapine in Non-Obese Mice

Hepatic and cardiac proteomes and serum immunomes from young adult C57BL/6 J mice were measured following oral treatment with risperidone (RIS, 1 mg/kg), olanzapine (OLAN, 5 mg/kg) or a drug vehicle (VEH, 0.1% acetic acid) for 4 weeks as previously described [9,16]. Briefly, 8-week-old male mice were fed a standard, low-fat chow diet for the duration of the study (28 days) and treated with RIS and OLAN doses, whose peak plasma drug concentrations fall within the range of plasma drug exposures observed clinically in patients. Hepatic and cardiac proteomes were determined by mass spectrometry and profiled by sequential window acquisition of all theoretical spectra (SWATH) analysis [16]. Serum immunomes were determined using the Proteome Profiler Mouse Cytokine Array (R&D Systems, Minneapolis, MN, USA). Differentially expressed (DE) proteins were identified by Student’s *t* test using a significance cutoff of *p* < 0.05. The proteomic datasets analyzed for this study can be found in the PeptideAtlas (identifier: PASS01349) http://www.peptideatlas.org/PASS/PASS01349 (Access date 20 January 2022). There were no significant changes in animal body weight, and NAFLD in AA-treated animals was histologically confirmed [7,9,16].

### 2.2. Transcriptomic Changes Induced by Risperidone Treatment of Human Neuroblastoma Cells In Vitro

Transcriptomes from low passage human neuroblastoma cells incubated in 24 h in RIS or the drug vehicle were generated as previously described [23]. Briefly, SH-SY5Y cells (pass 13) were cultured in Dulbecco’s modified Eagle medium supplemented with 10% (*v*/*v*) fetal bovine serum and either 100 nM RIS in dimethyl sulfoxide (DMSO) or DMSO alone. Total RNA was extracted and sequenced using the Illumina HiSeq platform. DE traits were identified with a threshold of log-fold change of 1.5 and a significance cutoff of *p* < 0.05. We accessed the DE dataset from entry GSE149611 in the NCBI GEO Database [27,28].

### 2.3. Transcriptomic Changes Induced by Onset of Aripiprazole Treatment in Psychiatric Patients

Transcriptional changes induced by aripiprazole (ARIP) treatment at first episode of psychosis were previously reported. Briefly, a randomized, flexible-dose, open-label study conducted at University Hospital Marques de Valdecilla, Santander (Spain) collected venous blood draws at patient intake and after three months of treatment with ARIP (5–30 mg/day) [29]. Total RNA was extracted and sequenced using the Illumina HiSeq platform. DE traits were identified by pairwise comparison of baseline transcriptomes to transcriptomes after 3 months of ARIP treatment using Student’s *t* test and using a significance cutoff of *p* < 0.05. We accessed the DE dataset from Appendix A, Crespo-Facorro et al. [30].

### 2.4. Compilation of Multi-Omic DE Traits

Four lists of DE traits were compiled from the assembled datasets: (1) DE traits in mice treated with OLAN (O_M_); (2) DE traits in mice treated with RIS (R_M_); (3) DE traits in human cells exposed to RIS (R_H_); (4) DE traits in psychiatric patients treated with ARIP (A_H_). Lists were examined for functional overlap among DE traits by pairing different subunits of multimeric proteins, identifying receptor/ligand pairs and identifying homologs and paralogs. DE traits that were common to more than one dataset were placed in distinct categories representing all factorial combinations as follows: O_M_R_M_, O_M_R_H_, O_M_A_H_, R_M_R_H_, R_M_A_H_, R_H_A_H_, O_M_R_M_R_H_, O_M_R_M_A_H_, R_M_R_H_A_H_ and O_M_R_M_R_H_A_H_.

### 2.5. Pathway Analysis to Detect Alterations in Traits Associated with NAFLD

The Database for Annotation, Visualization and Integrated Discovery (DAVID) v6.8 (https://david.ncifcrf.gov/home.jsp (accessed on 20 February 2022)) was used to query the Kyoto Encyclopedia of Genes and Genomes (KEGG) (https://www.genome.jp/kegg (accessed on 20 February 2022)) and Genetic Association Database (GAD) Disease databases with all 14 lists of DE traits and to assess involvement in the development of NAFLD [31,32,33,34]. Database accessions via DAVID occurred January 2022. Both the Functional Clustering and Functional Annotation Table tools were utilized. NAFLD, Inflammation, Cirrhosis, Obesity, Serum Lipid Levels, Body Mass Index, Lipid Metabolism, Bile Synthesis/Choline Secretion, Waist Circumference, Chronic Hepatitis and Thrombosis were identified as output categories and pathways significantly associated with the pathophysiology of NAFLD. Search terms allowing binning into each pathway are noted in Appendix A.

### 2.6. Pathway Analysis to Detect Alterations in Traits Associated with Iron Homeostasis

DAVID was also used to query the KEGG and GAD_Disease databases with each distinct list of DE traits to assess impacts on iron homeostasis. Iron, Hemoglobin, Hemochromatosis, Anemia, Iron Metabolism, Thalassemia and Blood Values were identified as output categories and pathways associated with iron homeostasis. Search terms allowing binning into each pathway are noted in Appendix A. Mechanistic figures depicting proposed changes in iron transport were generated using BioRender (www.BioRender.com (accessed on 20 February 2022)).

### 2.7. Measurement of Iron Accumulation in Livers of OLAN-Treated Mice

Iron species in livers from male mice treated with either VEH (*n* = 4) or OLAN (*n* = 5) for 28 days were analyzed for the presence of total iron (Fe), ferrous iron (Fe^2+^) or ferric iron (Fe^3+^) using a colorimetric iron assay (Iron assay kit #MAK025 Sigma Aldrich, St Louis, MO, USA). Briefly, a portion of liver (70–192 mg) from each animal was homogenized in assay buffer and adjusted to a final concentration of 250 mg/mL as determined by BCA protein assay (Novagen® Sigma-Aldrich, Saint Louis, MO, USA). These homogenates were centrifuged at 16,000× *g* for 10 minutes at 4 °C. The resulting supernatant (25 µL) was processed according to the manufacturer’s instructions. Absorbance was determined at 592 nm, and concentration of iron species was interpolated from a standard curve (2.00–10.0 nM) constructed from ferrous, ferric and total iron standards.

## 3. Results

### 3.1. Parameters of Subjects Assessed for Differential Expression Induced by Atypical Antipsychotic Drugs

Mice treated for four weeks with low-dose OLAN or RIS did not experience treatment effects on body weight, and all measured vital signs did not differ significantly between drug-treated animals and drug vehicle-treated control animals [7,16]. NAFLD was confirmed in OLAN and RIS mice by histological evaluation [7]. Psychiatric patients treated with ARIP had significantly elevated levels of serum prolactin and 23.6% experienced weight gain, as reported in a separate analysis of the same patient cohort [29]. Twenty-four-hour exposure of SH-SY5Y cells to 100 nM RIS did not result in cytotoxicity [27].

### 3.2. Multi-omic Analysis Predicts Impacts on NAFLD Development Following Atypical Antipsychotic Drug Exposure In Vitro and In Vivo

Pathway analysis utilizing both the KEGG and GAD_Disease Databases identified numerous DE traits associated with NAFLD or mechanistically associated clinical states. DE traits common to all factorial combinations of AAs and study systems were observed, as were DE traits unique to each dataset. Appendix A lists DE traits by dataset combinations and NAFLD-associated pathway. There were 85 NAFLD-associated DE traits common to human cells in vitro and human patients in vivo (Figure 2A), and 35 traits common to RIS exposure in vitro or in vivo (Figure 2B). Only 3 DE traits were found in all in vivo datasets and none in the in vitro (Figure 2C), whereas 40 NAFLD-associated DE traits were found across all four datasets (Figure 2D).

### 3.3. Multi-omic Analysis Predicts Impacts on Iron Homeostasis following Atypical Antipsychotic Drug Exposure In Vitro and In Vivo

Pathway analysis utilizing both the KEGG and GAD_Disease Databases identified numerous DE traits associated with iron homeostasis and anemia. Appendix A lists DE traits by dataset combinations and iron-associated pathway. There were 19 iron-associated DE traits common to human cells in vitro and human patients in vivo (Figure 3A), and 4 traits common to RIS exposure in vitro or in vivo (Figure 3B). A single DE trait, alipoprotein-1A, was found in all in vivo datasets and none were found in vitro (Figure 3C), whereas 7 iron-associated DE traits were found across all four datasets (Figure 3D). Traits unique to each dataset are noted in Figure 3E. The number of DE traits falling within GAD_Disease Database categories for anemia measures (i.e., hemoglobin, hematocrit, erythrocyte count and mean corpuscular volume) are shown for each in vivo system (Figure 4A). The direction of multiple DE traits led the authors to develop a proposed mechanism for the development of clinical anemia following AA exposure (Figure 4B).

### 3.4. Proposed Mechanisms of Altered Iron Homeostasis Following Pathway Analysis of Traits Differentially Expressed by Atypical Antipsychotic Drugs

Pathway analysis utilizing both the KEGG and GAD_Disease Databases identified numerous DE traits associated with iron homeostasis. The directional change of multiple DE traits led to the development of a proposed mechanism for altered cellular iron uptake following AA exposure, and how this impairment predicts reduction in circulating ferritin and accumulation of iron in tissues (Figure 5).

### 3.5. Measurement of Accumulated Iron in Mice Treated with OLAN

In order to test our hypothesis that AA-associated NAFLD is due, at least in part, to drug-induced changes in iron homeostasis, we quantified the concentration of iron species in livers collected from male mice treated sub-chronically with OLAN (Figure 6). We previously reported that these mice developed drug-associated lean NAFLD as determined by histological evaluation [7]. Clinically relevant exposure of AA caused a significant (*p* < 0.009) increase in total Fe concentrations in the liver consistent with gene expression changes in pathways associated with regulation of iron homeostasis coincident with NAFLD.

### 3.6. Proposed Mechanisms of NAFLD Development Following Pathway Analysis of Traits Differentially Expressed by Atypical Antipsychotic Drugs

Pathway analysis utilizing both the KEGG and GAD_Disease Databases identified numerous DE traits associated with NAFLD and iron homeostasis. The directional change of multiple DE traits led to the development of a proposed mechanism for AA-induced NAFLD (Figure 7).

## 4. Discussion

Development of NAFLD has been associated with AA treatment in preclinical models and patient populations [35,36,37,38], and patients treated with AA medications have been reported to have significantly higher rates of iron-deficiency anemia [17,26,39,40,41,42,43,44] and obesity [12,13,14,15]. Despite these reports, pathophysiological mechanisms by which AA treatment can lead to NAFLD and disruptions in iron homeostasis remain virtually unexplored. We combined multi-omic data sets utilizing three different AA medications, one in vitro model system, two in vivo preclinical model systems and one patient cohort to identify commonalities across drugs and study systems in order to propose potential mechanisms leading to these complications that can be prospectively evaluated in future studies wherein changes to specific traits are measured. Whereas NAFLD could only be histologically confirmed in the two preclinical models, the inclusion of data from subjects with and without drug-associated increase in body weight (i.e., lean preclinical animals and a patient cohort with reported weight gain at 3 months of drug treatment) indicates that the data from the in vivo systems represent a spectrum of AA-induced metabolic effects. Importantly, our multi-omic approach provided preliminary evidence that drug effects on metabolic dysregulation as observed in NAFLD is translational across the drug class and across species and model systems. Furthermore, dysregulation of gene and protein expression in NAFLD-associated pathways occurs at the cellular and organismal level, in the presence and absence of obesity and with acute and subchronic treatment duration. We note that NAFLD was only histologically confirmed in pre-clinical models, and mechanistic overlap with in vitro and in vivo gene expression changes should be considered supportive of mechanistic hypotheses rather than trans-species evidence of proposed mechanisms. Prospective evaluations of potential biomarkers in humans require further evaluation. However, screening guidelines for NAFLD recently released by the American Association of Clinical Endocrinology recommend calculation of a fibrosis-4 (FIB-4) index for high-risk patients [45] in primary care or endocrinology outpatient settings. This is noteworthy as platelet levels are a crucial component of FIB-4 index calculations, and our omic data-driven mechanistic hypotheses predict changes to risk of thrombosis as shown by DE of 15 thrombosis-related traits across all datasets (Appendix A).

The liver plays a central role in the regulation of whole-body iron homeostasis, and chronic liver diseases such as NAFLD and NASH are associated with disruption in iron metabolism [46,47]. Reduced liver function in NAFLD/NASH can lead to hepatic iron overload, culminating in oxidative stress and iron-associated cellular damage [48]. Here, we report that three AA medications alter multiple pathways implicated in the regulation of iron metabolism across model systems, and in the presumed presence (psychiatric patients) or absence (mice) of obesity. We confirm AA treatment altered total hepatic iron content in the livers from mice treated with OLAN (Figure 6) with coincident histological evidence of NAFLD as previously published [7]. The relationship between NAFLD, hemoglobin levels and erythrocyte count is becoming increasingly appreciated to the extent that both erythrocyte count and hemoglobin level have been proposed as potential biomarkers for NAFLD [49,50].

Our proposed model of impaired iron transport (Figure 5) involves a decrease in cation channels (CACNA proteins), decreased transferrin co-receptor HFE and impaired vesicle formation indicating a reduction in cellular iron and an accumulation of extracellular iron. Whereas CACNA proteins were originally described as calcium channels, they have been strongly implicated in iron transport as well [51]. The reduction of intracellular iron leads to impairment of mitochondrial function and decreased ferritin-coupled iron. We and others have reported AA-associated mitochondrial dysfunction as an underlying cause of AA-associated metabolic side effects [16,52]. Hypoferritinemia and/or iron-deficiency anemia has also been reported for patients treated with risperidone [21,26,39,40]. Additionally, chronic treatment (12 weeks) of rats with the AA medication clozapine or the first generation (typical) antipsychotic medication haloperdol resulted in sex-dependent effects on hepatic iron metabolism and anemia [53]. Sudden-onset anemia associated with clozapine and lurasidone that resolves upon cessation of treatment has also been described in case reports [41,42]. These data, which were generated using antipsychotic medications with distinct pharmacology from the AA drugs we report, are nevertheless consistent with our findings for effects of the atypical antipsychotic drugs olanzapine, risperidone and aripiprazole on hepatic iron metabolism. Taken together, these findings support our hypothesis mechanistically linking disrupted iron homeostasis, AA-induced anemia and NAFLD to AA medications. Future prospective testing of these hypotheses will be critical to the development of biomarkers for the clinical evaluation of NAFLD in addition to screening by FIB-4 index.

## 5. Conclusions

Incidence of NAFLD is increased in patients with psychiatric illness, and antipsychotic medications independently increase NAFLD risk. AAs are widely prescribed to treat a variety of psychiatric disorders and the increase in their off-label use has raised reasonable concerns regarding their significant metabolic side effects. In this paper, we employed a multi-omic, mechanistic approach to better understand the mechanistic pharmacology of AA-induced metabolic disease, including NAFLD and anemia. We found AA-associated dysregulation of NAFLD-associated pathways linking insulin resistance, inflammation, mitochondrial dysfunction and altered iron metabolism in the absence and presence of drug-associated weight gain. These novel findings highlight the significant impact of AA medications on metabolic burden and the need for holistic patient monitoring beyond a focus on weight gain. Clinical guidelines for metabolic monitoring of patients taking AA should be closely followed, especially in vulnerable patient populations, including children [54]. Specifically, monitoring baseline and sustained bodyweight, waist circumference, blood pressure and biomarkers relating to glycemia, lipids and liver function (enzymes), and changes to FIB-4 index are recommended. Clinical monitoring guidelines specifically focused on NAFLD/NASH in psychiatry have not yet been developed.

## Figures and Tables

**Figure 1 biomedicines-10-01225-f001:**
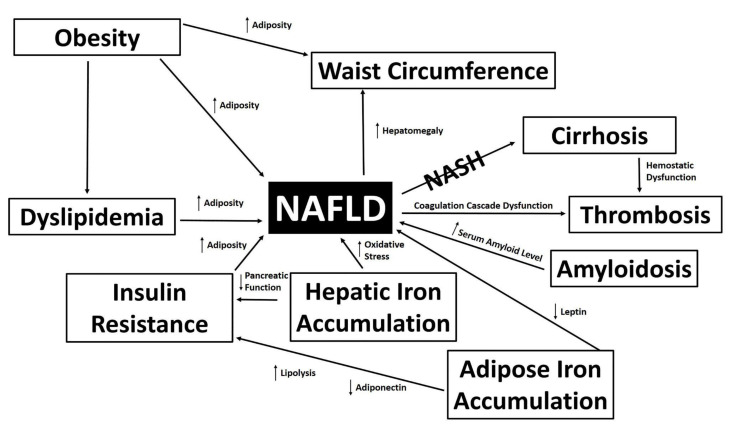
The Pathophysiology of NAFLD. Clinical states and changing metrics associated with NAFLD are depicted.

**Figure 2 biomedicines-10-01225-f002:**
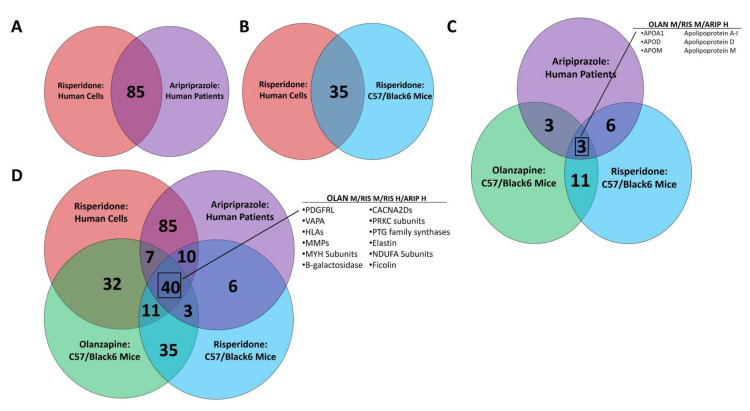
Differentially Expressed Traits Commonly Associated with NAFLD. Venn diagrams representing overlap between AA-exposed multi-omic datasets were generated. Transcriptomic changes in psychiatric patients treated with ARIP are shown in pink, and in human neuroblastoma cells exposed to RIS are shown in purple. Proteomic changes in non-obese mice treated with OLAN are shown in green, and in those treated with RIS are shown in blue. Common changes in human systems in vitro or in vivo are shown, (**A**) as are common changes induced by RIS in vitro and in vivo. (**B**) The number of changes seen in vivo that are not seen in vitro are shown (**C**). DE traits common across study systems are shown in Panel (**D**), and those of notable interest are listed.

**Figure 3 biomedicines-10-01225-f003:**
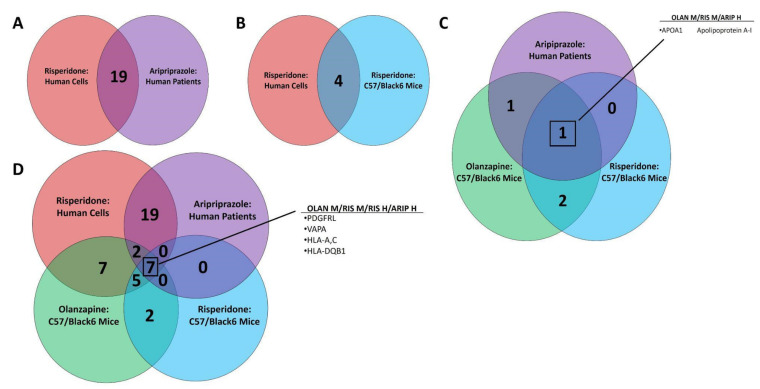
Common Iron-Associated Differentially Expressed Traits. Venn diagrams representing overlap in DE traits related to iron homeostasis between AA-exposed multi-omic data sets were generated. Transcriptomic changes in psychiatric patients treated with ARIP are shown in pink, and in human neuroblastoma cells exposed to RIS are shown in purple. Proteomic changes in non-obese mice treated with OLAN are shown in green, and in non-obese mice treated with RIS are shown in blue. Common changes in human systems in vitro or in vivo are shown, (**A**) as are common changes induced by RIS in vitro and in vivo. (**B**) The number of changes seen in vivo that are not seen in vitro are shown. (**C**) DE traits common across study systems are shown in Panel (**D**), and the most critical traits are noted.

**Figure 4 biomedicines-10-01225-f004:**
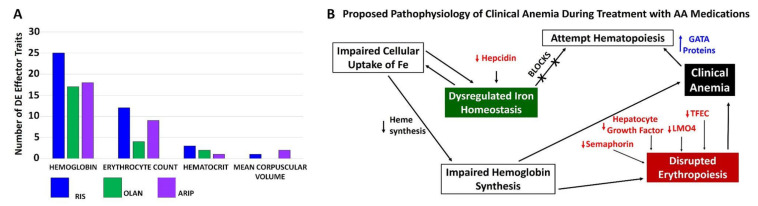
Differentially Expressed Traits In Vivo and Development of Clinical Anemia. Total numbers of DE traits associated with clinical measures of anemia (hemoglobin, hematocrit, erythrocyte count and mean corpuscular volume) in mice treated with RIS (panel (**A**), blue bars), mice treated with OLAN (panel (**A**), green bars) and patients treated with ARIP (panel (**A**), purple bars) are shown. The directional change of multiple DE traits led to the development of a proposed mechanism for the development of clinical anemia following AA exposure (**B**).

**Figure 5 biomedicines-10-01225-f005:**
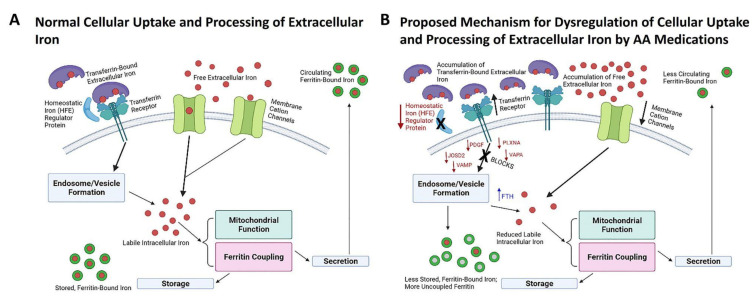
Proposed Mechanism for Impaired Cellular Iron Uptake Based on Differentially Expressed Traits. Panel (**A**) schematically describes processes for cellular iron uptake, storage and secretion into the circulatory system under physiologically normal conditions. AA-induced decreases of membrane cation channels, the transferrin co-receptor HFE and numerous proteins involved in endosome and vesicle formation suggests an inability to efficiently transport iron into cells and result in accumulation of extracellular iron in the liver (Panel (**B**)). The reduction in intracellular iron would impair mitochondrial function and reduce the amount of available ferritin-bound iron within cells and in circulation.

**Figure 6 biomedicines-10-01225-f006:**
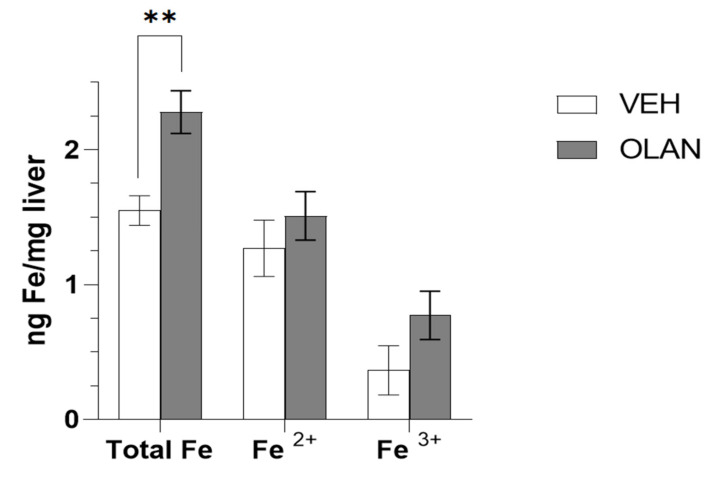
Iron (Fe) concentration in livers of mice treated with vehicle (VEH) or olanzapine (OLAN). Concentrations of iron species were determined in mice treated with VEH (*n* = 4) or OLAN (*n* = 5) daily for 28 days (5 mg/kg OLAN). ** *p* < 0.009.

**Figure 7 biomedicines-10-01225-f007:**
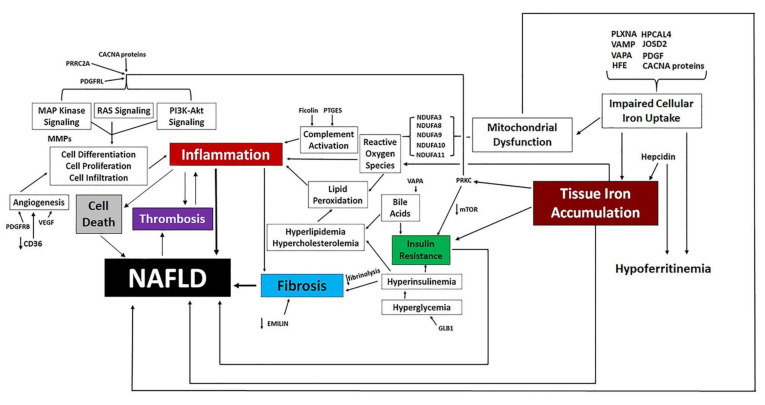
Proposed Molecular Mechanisms Underlying the Clinical Development of AA-Induced NAFLD.

## Data Availability

Original datasets are available as described in [7,9,12,23,25] and as noted in the Methods. Complete lists of DE traits commonly expressed between groups can be found at (DOI: 10.13140/RG.2.2.20998.65601).

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
