# Peer review of "Mechanisms Underlying Antipsychotic-Induced NAFLD and Iron Dysregulation: A Multi-Omic Approach"

_biomedicines, 2022, doi:10.3390/biomedicines10061225_

Round 1

Reviewer 1 Report

Title: Mechanisms Underlying Antipsychotic Induced NAFLD: A Multi-omic Approach

Authors: Meghan May, Deborah Barlow, Radwa Ibrahim and Karen L. Houseknecht

General Comment:

The metabolic complications, such as non-alcoholic fatty liver disease (NAFLD), constitute a significant limitation in using atypical antipsychotics (AA) medications. Understanding the molecular mechanisms leading to the development of these phenomena may provide an opportunity to design new therapeutic strategies to reduce the risk of metabolic complications associated with AA treatment. In their work, Meghan May et al., using a multi-omic approach, investigated the pathogenesis of NAFLD accompanying AA treatment, finding the molecular link between the AA-induced disturbance of iron metabolism and the development of liver steatosis. The idea of the work is interesting, and the findings would constitute a significant contribution to the development of the field; however, some methodological issues should be clarified before the manuscript is accepted for publication.

Major revisions:

Title:  

The main finding of the work refers to the link between the AA treatment-induced impairment of iron metabolism and NAFLD development. This information should be included in the manuscript title as well.

Methodology and Results

The work combines the results of several studies of different experimental designs (in vitro, in vivo, and a human study) and uses different active compounds. The influence of AA treatment on gene expression analysis was once performed on mRNA, once one protein level in different spectra of biological material (murine heart and liver tissue, human neuroblastoma cell line, human sera). Apart from the animal study, where liver histology and iron content in the livers were assessed, no other clinical outcomes related to iron metabolism and liver structure/function were evaluated. Contrary, the primary outcomes of the in vitro study referred to drug-induced cytotoxicity, while in the human study – the influence of AA treatment on the prolactin level and weight gain was assessed. Therefore the proposed models linking AA treatment with the disturbed iron metabolism and NAFLD development, based on the bioinformatic algorithms, should be treated with caution.

To increase the reliability of the findings, the Authors should limit the analysis to the studies where results of AA administration were verified by hepatic outcomes.

Discussion:

The authors should discuss some limitations of the study (e.g., the heterogeneity of the experimental models used for the bioinformatic analyses).

Author Response

 Reviewer 1

The main finding of the work refers to the link between the AA treatment-induced impairment of iron metabolism and NAFLD development. This information should be included in the manuscript title as well.

  • The original title, “Mechanisms Underlying Antipsychotic Induced NAFLD: A Multi-omic Approach” was revised to: “Mechanisms Underlying Antipsychotic Induced NAFLD and Iron Dysregulation: A Multi-omic Approach”

Apart from the animal study, where liver histology and iron content in the livers were assessed, no other clinical outcomes related to iron metabolism and liver structure/function were evaluated. Contrary, the primary outcomes of the in vitro study referred to drug-induced cytotoxicity, while in the human study – the influence of AA treatment on the prolactin level and weight gain was assessed. Therefore the proposed models linking AA treatment with the disturbed iron metabolism and NAFLD development, based on the bioinformatic algorithms, should be treated with caution.

  • The following language was added to Discussion Paragraph 1: “We note that NAFLD was only histologically confirmed in pre-clinical models, and mechanistic overlap with in vitro and in vivo gene expression changes should be considered supportive of mechanistic hypotheses rather than trans-species evidence of proposed mechanisms. Prospective evaluations of potential biomarkers in humans require further evaluation. However, screening guidelines for NAFLD recently released by the American Association of Clinical Endocrinology recommend calculation of a fibrosis-4 (FIB-4) index for high risk patients [45] in primary care or endocrinology outpatient settings. This is noteworthy as platelet levels are a crucial component of FIB-4 index calculations, and our omic data-driven mechanistic hypotheses predict changes to risk of thrombosis as shown by DE of 15 thrombosis-related traits across all datasets (see Table S3).”

The authors should discuss some limitations of the study (e.g., the heterogeneity of the experimental models used for the bioinformatic analyses).

  • The following language was added to Discussion Paragraph 1: “We combined multi-omic data sets utilizing three different AA medications, one in vitro model system, two in vivo preclinical model systems, and one patient cohort to identify commonalities across drugs and study systems in order to propose potential mechanisms leading to these complications that can be prospectively evaluated in future studies wherein changes to specific traits are measured.”
  • The following language from Discussion Paragraph 1, “our multi-omic approach provided robust evidence that drug effects on metabolic dysregulation” was revised to: “our multi-omic approach provided preliminary evidence that drug effects on metabolic dysregulation”

Reviewer 2 Report

This work presents an interesting approach in assessing the link between AA exposure, iron metabolism and NAFLD on various types of preclinical and clinical experimental samples. 
However, I believe that it is necessary to better investigate the possible interconnections between the results obtained and the possible clinical implications. For example, through what mechanisms can AA lead to the observed alterations? What kind of studies could be envisaged in the future to better undestand these unwanted effects of AAs? What might be the take home message for psychiatrists, endocrinologists, and hepatologists? 
I also point out two specific comments: 
1) (paragraph 2.3) was a correction made for multiple comparisons?
2) (second paragraph of the discussion) why is it stated that the obesity condition is present in patients with mental disorders? 

Author Response

Reviewer 2

I believe that it is necessary to better investigate the possible interconnections between the results obtained and the possible clinical implications. For example, through what mechanisms can AA lead to the observed alterations?

  • Figures 4, 5 and 7 were designed to help illustrate connections between mechanistic changes observed across the datasets (multi-omic, multi-species approach) and the manifestation of clinical NAFLD in patients with mental illness that has been widely reported in the literature. We have revised the title of Figure 7 to reflect the holistic mechanistic view (proteomics to pathophysiology and clinical presentation of disease). Furthermore Figure 4 illustrates mechanistic hypothesis for how observed results and development of Clinical Anemia. Figure 5 further illustrates this hypothesis with integration of drug-associated changes in Fe metabolism linked to literature-reported clinical outcomes.

What kind of studies could be envisaged in the future to better understand these unwanted effects of AAs?

  • The following language was added to Discussion Paragraph 1: “We combined multi-omic data sets utilizing three different AA medications, one in vitro model system, two in vivo preclinical model systems, and one patient cohort to identify commonalities across drugs and study systems in order to propose potential mechanisms leading to these complications that can be prospectively evaluated in future studies wherein changes to specific traits are measured.”

What might be the take home message for psychiatrists, endocrinologists, and hepatologists? 

  • The following language was added to Discussion Paragraph 3: “Future prospective testing of these hypotheses will be critical to the development of biomarkers for the clinical evaluation of NAFLD in addition to screening by FIB-4 index.”

  • The following language was added to the Conclusion section: “ Clinical guidelines for metabolic monitoring of patients taking AA should be closely followed, especially in vulnerable patient populations including children [54-55].  Specifically, monitoring baseline and sustained bodyweight, waist circumference, blood pressure and biomarkers relating to glycemia, lipids and liver function (enzymes), and changes to FIB-4 index are recommended. Clinical monitoring guidelines specifically focused on NAFLD/NASH in psychiatry have not yet been developed.”   

Paragraph 2.3: was a correction made for multiple comparisons?

  • This analysis compared the expression level of each individual trait prior to treatment and at a single time point post treatment. No comparisons were made between traits or across >2 time points; therefore, multiple comparisons were not made.

Why is it stated that the obesity condition is present in patients with mental disorders? 

  • The following language was added to Introduction Paragraph 1: “Rapid weight gain is a common occurrence in first-episode psychosis treated with AA medications [12-14] , and the majority of patients have persistent body mass indices (BMI) classified as obese [15].”

  • The following language from Discussion Paragraph 2, “Here we report that 3 AA medications alter multiple pathways implicated in the regulation of iron metabolism across model systems, and in the presence (psychiatric patients) or absence (mice) of obesity” was revised to ““Here we report that 3 AA medications alter multiple pathways implicated in the regulation of iron metabolism across model systems, and in the presumed presence (psychiatric patients) or absence (mice) of obesity”.

  • The following language from Discussion Paragraph 1, “…patients treated with AA medications have been reported have significantly higher rates of iron-deficiency anemia [17, 22, 35-39]” was revised to “…patients treated with AA medications have been reported have significantly higher rates of iron-deficiency anemia [17, 22, 35-39] and obesity [12-15].”

Round 2

Reviewer 1 Report

I want to express my gratitude for the opportunity to re-review the paper entitled: "Mechanisms Underlying Antipsychotic Induced NAFLD: A Multi-omic Approach" by Meghan May et al. Since the authors addressed my concerns regarding the methodology, I find the manuscript acceptable for publication in Biomedicines.

Reviewer 2 Report

I thank the authors for the changes made to the manuscript.